# OsWRKY5 Promotes Rice Leaf Senescence via Senescence-Associated NAC and Abscisic Acid Biosynthesis Pathway

**DOI:** 10.3390/ijms20184437

**Published:** 2019-09-09

**Authors:** Taehoon Kim, Kiyoon Kang, Suk-Hwan Kim, Gynheung An, Nam-Chon Paek

**Affiliations:** 1Department of Plant Science, Plant Genomics and Breeding Institute, Research Institute of Agriculture and Life Sciences, Seoul National University, Seoul 08826, Korea; 2Crop Biotech Institute and Graduate School of Biotechnology, Kyung Hee University, Yongin 17104, Korea

**Keywords:** rice, leaf senescence, abscisic acid (ABA), OsWRKY, NAC

## Abstract

The onset of leaf senescence is triggered by external cues and internal factors such as phytohormones and signaling pathways involving transcription factors (TFs). Abscisic acid (ABA) strongly induces senescence and endogenous ABA levels are finely tuned by many senescence-associated TFs. Here, we report on the regulatory function of the senescence-induced TF OsWRKY5 TF in rice (*Oryza sativa*). *OsWRKY5* expression was rapidly upregulated in senescing leaves, especially in yellowing sectors initiated by aging or dark treatment. A T-DNA insertion activation-tagged *OsWRKY5*-overexpressing mutant (termed *oswrky5-D*) promoted leaf senescence under natural and dark-induced senescence (DIS) conditions. By contrast, a T-DNA insertion *oswrky5*-knockdown mutant (termed *oswrky5*) retained leaf greenness during DIS. Reverse-transcription quantitative PCR (RT-qPCR) showed that *OsWRKY5* upregulates the expression of genes controlling chlorophyll degradation and leaf senescence. Furthermore, RT-qPCR and yeast one-hybrid analysis demonstrated that OsWRKY5 indirectly upregulates the expression of senescence-associated *NAM/ATAF1/2/CUC2* (*NAC*) genes including *OsNAP* and *OsNAC2*. Precocious leaf yellowing in the *oswrky5-D* mutant might be caused by elevated endogenous ABA concentrations resulting from upregulated expression of ABA biosynthesis genes *OsNCED3*, *OsNCED4*, and *OsNCED5*, indicating that OsWRKY is a positive regulator of ABA biosynthesis during leaf senescence. Furthermore, *OsWRKY5* expression was suppressed by ABA treatment. Taken together, *OsWRKY5* is a positive regulator of leaf senescence that upregulates senescence-induced *NAC*, ABA biosynthesis, and chlorophyll degradation genes.

## 1. Introduction

Leaf senescence is the final stage of plant development and involves diverse molecular and cellular processes such as degradation of chlorophylls and macromolecules, and remobilization of nutrients into newly developing or storage organs through expression of senescence-associated genes (SAGs). The onset of leaf senescence begins with chlorophyll degradation and proceeds to hydrolysis of macromolecules (proteins, lipids, and nucleic acids); this is followed by cell death [1,2,3].

Genetic studies have revealed the contribution of chlorophyll catabolic enzymes to sequential reactions of chlorophyll degradation. The STAY-GREEN (SGR) protein is a magnesium (Mg)-dechelatase, which produces pheophytin *a* by removing Mg from chlorophyll *a* [4,5]. Thus, functional deficiency of SGR orthologs leads to a strong stay-green phenotype in diverse plant species including *Arabidopsis thaliana* [6], rice (*Oryza sativa*) [4], pea (*Pisum sativum*) [7], tomato (*Solanum lycopersicum*), bell pepper (*Capsicum annuum*) [8], and soybean (*Glycine max*) [9]. Failure to convert pheophytin *a* to pheophorbide *a* due to mutation of *NON-YELLOW COLORING3* (*NYC3*), encoding an α/β hydrolase-fold family protein, delays leaf senescence during dark-induced senescence (DIS) [10]. Knockdown of rice *pheophorbide a oxygenase* (*OsPAO*) leads to accumulation of pheide *a* and prolongs leaf greenness during dark incubation [11]. SAGs identified during leaf senescence in rice encode putative proteins involved in metabolic programing [12]; *Osh36* and *Osl85* encode an aminotransferase and isocitrate lyase, which participate in amino acid and fatty acid metabolism, respectively.

Abscisic acid (ABA) participates in multiple aspects of plant development including leaf senescence, seed germination, stomatal closure, and root development [13,14,15,16,17,18]. Specifically, expression of ABA biosynthetic genes such as those encoding 9′-*cis*-epoxycarotenoid dioxygenases (NCEDs) is induced by leaf senescence, elevating endogenous ABA levels in Arabidopsis leaves [19,20]. Increased levels of endogenous ABA can activate chlorophyll degradation pathways mediated by senescence-associated transcription factors (TFs) [21,22,23,24]. For instance, ABA induces the expression of the genes encoding ABA-RESPONSIVE ELEMENT (ABRE)-BINDING TRANSCRIPTION FACTOR 2 (ABF2), ABF3, and ABF4, which directly bind to the *SGR1* promoter, accelerating chlorophyll degradation in Arabidopsis leaves [21]. In rice, ABA-promoted expression of *OsNAP* directly upregulates chlorophyll degradation genes (CDGs) such as *SGR*, *NYC1*, *NYC3*, and *RCCR1*, leading to early leaf senescence [25].

The WRKY TFs participate in various biological processes such as biotic and abiotic stress, seed development, seed dormancy, and germination [26]. Genome-wide analyses have revealed that many WRKY genes are strongly induced by leaf senescence [27,28], suggesting that WRKY TFs are involved in regulating leaf senescence. Following identification of AtWRKY6 as a regulator of leaf senescence [29], other WRKY TFs regulating leaf senescence have been functionally characterized. For example, mutation of Arabidopsis *AtWRKY53* confers a delayed leaf senescence phenotype by specifically altering regulation of its target genes [30]. Overexpression of *AtWRKY22*, a target gene of AtWRKY53, accelerates leaf senescence [31]. AtWRKY54 and AtWRKY70 act as negative regulators of leaf senescence by interacting independently with WRKY30 [32]. AtWRKY45 mediates gibberellic acid (GA)-induced leaf senescence by interacting with a DELLA protein, RGL1 [33]. AtWRKY75 increases salicylic acid (SA) and H_2_O_2_ levels by activating *SID2* and repressing *CAT2*, respectively, resulting in early leaf senescence [34]. Heterologous expression of rice *OsWRKY23* promotes leaf senescence in Arabidopsis [35]. Rice OsWRKY42 induces reactive oxygen species (ROS) by directly downregulating the expression of *OsMT1d* encoding metallothionein protein and thereby promoting leaf senescence [36]. Unlike Arabidopsis WRKY TFs involved in the regulation of leaf senescence; however, few OsWRKY TFs have been identified as functioning in the execution of leaf senescence.

In this study, we found that *OsWRKY5* expression is upregulated at the onset of leaf senescence. The *OsWRKY5*-overexpressing *oswrky5-D* mutation promoted leaf yellowing under aging and dark treatment, while an *oswrky5*-knockdown mutant exhibited a delayed senescence phenotype during DIS. Reverse-transcription quantitative PCR (RT-qPCR) analysis suggested that CDGs and SAGs were upregulated by senescence-induced OsWRKY5. Furthermore, OsWRKY5 seemed to indirectly regulate the expression of senescence-associated *NAC* (*senNAC*) genes such as *OsNAP* and *OsNAC2*, which are upstream regulators of CDGs and SAGs. *OsWRKY5* elevated endogenous ABA levels by upregulating the expression of ABA biosynthetic genes. Our results thus provide evidence that OsWRKY5 acts as a positive regulator of leaf senescence in rice.

## 2. Results

### 2.1. Characterization of OsWRKY5

*OsWRKY5* (Os05g04640), a member of rice WRKY TF family, consists of six exons with 1509 bp of open reading frame in 5379 bp of genomic DNA. *OsWRKY5* is predicted to encode a 502 amino acid protein with a molecular mass of 52.3 kDa (http://rice.plantbiology.msu.edu/index.shtml). The WRKY domain of OsWRKY5 has a single consensus motif (WRKYGQK) and a zinc-finger C_2_H_2_ motif (Cx_5_Cx_23_HxH), indicating that OsWRKY5 belongs to the group II WRKY TF family [37]. From sequence alignment of WRKY domains between OsWRKY5 and group II *Arabidopsis thaliana* WRKY (AtWRKY) proteins, we found that the domain sequences of OsWRKY5 are quite similar to those of AtWRKY6 and AtWRKY47, members of the subgroup IIb AtWRKY TF family (Appendix A). To examine the subcellular localization of OsWRKY5, we transiently expressed the *35S::YFP-OsWRKY5* construct in onion epidermal cells. The fluorescent signal of YFP-OsWRKY5 fusions was observed exclusively in nuclei (Appendix A), indicating that OsWRKY5 is a nuclear-localized protein.

### 2.2. OsWRKY5 Is Upregulated during Leaf Senescence

To examine the spatial expression of *OsWRKY5*, we investigated transcript levels of *OsWRKY5* in rice organs including root, culm, leaf blade, leaf sheath, and panicle at the reproductive stage (Figure 1a). *OsWRKY5* was preferentially expressed in the leaf blade and leaf sheath. Previous transcriptome analysis [28] showed upregulation of 47 rice WRKY TFs including *OsWRKY5* in flag leaves during natural senescence (NS). Therefore, we determined age-dependent changes in *OsWRKY5* expression in flag leaves of wild-type (WT; *japonica* cultivar ‘Dongjin’) rice grown in a paddy field under natural long-day (NLD) conditions (>14 h light/day). While *OsWRKY5* was constitutively expressed in developing leaves at the vegetative stage, *OsWRKY5* expression was dramatically upregulated in senescing leaves at the reproductive stage (Figure 1b). In addition, *OsWRKY5* expression gradually increased in detached leaves of four-week-old WT leaves during DIS (Figure 1c). We further found that *OsWRKY5* transcripts accumulated in the yellowing sector (region c) more than in the green sector (region a) of senescing flag leaves (Figure 1d). These results suggest that *OsWRKY5* is involved in the onset and progression of leaf senescence in rice.

### 2.3. OsWRKY5 Positively Regulates the Progression of Leaf Senescence

To examine the function of *OsWRKY5* in leaf senescence, we identified activation-tagged and loss-of-function mutants. To this end, we obtained two independent T-DNA insertion lines (PFG_3A-15928 and PFG_3A-06060) from the RiceGE database (http://signal.salk.edu/cgi-bin/RiceGE) in which each T-DNA fragment with an activation tag (4 × 35S promoter) was integrated into the promoter region of *OsWRKY5* (Figure 2a). Nucleotide sequences used for determining the locations of T-DNA insertions have been posted on the RiceGE database. Based on the comparison with *OsWRKY5* promoter sequence, we predicted that T-DNA fragments of *oswrky5-D* and *oswrky5* are inserted in the 1602-bp and 1637-bp upstream of the start codon of OsWRKY5 gene, respectively. They were confirmed by our re-sequencing the regions of T-DNA insertion regions of two mutant lines. To verify the expression levels of *OsWRKY5* in these mutant lines, we measured *OsWRKY5* expression levels in detached leaves of four-week-old mutant plants during DIS. RT-qPCR showed that *OsWRKY5* transcripts accumulated to high levels in PFG_3A-15928 compared to the WT due to the activation-tagged T-DNA insertion; by contrast, in PFG_3A-06060, the T-DNA insertion in the promoter region of *OsWRKY5* reduced expression of *OsWRKY5* (Figure 2b,c). These results indicate that PFG_3A-15928 and PFG_3A-06060 are dominant activation and recessive knockdown mutants, respectively (hereafter termed *oswrky5-D* and *oswrky5*, respectively). To confirm this, we further investigated the *OsWRKY5* expression in leaf blade, leaf sheath, and root separated from WT and mutant lines grown in paddy soil for three weeks. Similar to expression patterns of *OsWRKY5* shown in detached leaves during DIS (Figure 2b,c), *OsWRKY5* transcripts highly accumulated in all tissues of *oswrky5-D* compared with the WT, while they were significantly lower in *oswrky5* than in the WT (Figure 2d,e).

To determine the phenotypic difference between WT and mutant lines during DIS, we next incubated detached leaves of three-week-old WT, *oswrky5-D*, and *oswrky5* plants in 3 mM MES buffer (pH 5.8) at 28 °C under complete darkness. While *oswrky5-D* showed accelerated leaf yellowing compared with the WT, the *oswrky5* leaves retained their green color longer than the WT leaves (Figure 3a,b). Consistent with the leaf color, the total chlorophyll content of *oswrky5-D* was less than that of the WT after DIS, whereas *oswrky5* maintained higher total chlorophyll levels during DIS compared with the WT (Figure 3c,d).

In senescing leaves, chlorophylls are sequentially degraded by chlorophyll-degrading enzymes associated with upregulation of CDGs, including SGR [4], NYC3 [10], and OsPAO [11]. Many other SAGs are also upregulated during DIS in rice, with products identified as seed imbibition protein (Osh69), glyoxylate aminotransferase (Osh36), and isocitrate lyase (Osl85) [12]. We therefore measured transcript levels of CDGs and SAGs in detached leaves of three-week-old WT, *oswrky5-D*, and *oswrky5* plants under DIS conditions as shown in Figure 2. RT-qPCR analysis revealed that expression of CDGs and SAGs was upregulated in *oswrky5-D* after 4 days of dark incubation (DDI) (Figure 4a–f) but downregulated in *oswrky5* after 5 DDI when compared with the WT (Figure 4g–l). These results demonstrate that *OsWRKY5* promotes the onset and progression of leaf senescence by upregulating expression of CDGs and SAGs.

To further examine how *OsWRKY5* overexpression affects leaf senescence during vegetative and reproductive stages, we monitored age-dependent leaf yellowing in WT and *oswrky5-D* plants grown under NLD conditions (>14 h daylight) in the field (37° N latitude, Suwon, South Korea). While there was no significant difference in leaf color between the WT and *oswrky5-D* until heading (Figure 5a), the leaves of *oswrky5-D* showed a precocious leaf senescence phenotype at 40 days after heading (DAH) (Figure 5b,c). The SPAD value, a parameter for leaf greenness, indicated lower levels of green pigments in flag leaves of *oswrky5-D* compared with the WT at 24 DAH (Figure 5d). Leaf greenness is closely linked to photosynthetic capacity [38,39]. Thus, reduced SPAD value led to a relatively lower *Fv/Fm* ratio (efficiency of photosystem II) in *oswrky5-*D than in WT at 24 DAH (Figure 5e). Similar to expression patterns of CDGs and SAGs during DIS, CDG and SAG transcripts were more abundant in the senescing flag leaves of *oswrky5-D* than in those of WT at 40 DAH (Figure 5f). These results indicate that OsWRKY5 acts as a positive regulator of leaf senescence during both NS and DIS.

### 2.4. OsWRKY5 Upregulates SenNAC Genes

Previous studies have shown that senNACs including OsNAP and OsNAC2 promote leaf senescence by upregulating expression of CDGs and SAGs [25,40]. To determine whether *OsWRKY5* participates in NAC TF-mediated senescence pathways, we examined the expression levels of *OsNAP* and *OsNAC2* in detached leaves of WT, *oswrky5-D*, and *oswrky5* under DIS conditions. RT-qPCR showed that compared with WT, the expression levels of *OsNAP* and *OsNAC2* were higher in *oswrky5-D* during dark incubation (Figure 6a,b), while they were downregulated at 0 and 5 DDI in *oswrky5* compared with WT (Figure 6c,d). These results suggest that *OsWRKY5* promote leaf senescence by upregulating the expression of *OsNAP* and *OsNAC2*.

WRKY TFs regulate the transcription of their target genes by recognizing a consensus *cis*-element, the so-called W-box [26]. The W-box has been generally defined as 5′-TTGAC(C/T)-3′ with an invariant TGAC core sequence essential for WRKY binding [37,41]. Since repetitive TGAC sequences enhance WRKY binding efficiency, we searched for the TGAC core sequence within 2 kb upstream of the transcriptional initiation sites of *OsNAP* and *OsNAC2*, and found two regions (‒1001 ~ ‒765 and ‒657 ~ ‒575) in the promoter of *OsNAP* and five regions (‒1760 ~ ‒1574, ‒1411 ~ ‒1309, ‒1135 ~ ‒1021, ‒660 ~ ‒480, and ‒352 ~ ‒98) in the promoter of *OsNAC2* (Figure 6e). To investigate whether the OsWRKY5 TF directly binds to the promoters of *OsNAP* and *OsNAC2*, we performed yeast one-hybrid assays. However, we could not find any difference between GAL4AD and GAL4AD-OsWRKY5 by measuring β-galactosidase activity of *lacZ* reporter genes, indicating that OsWRKY5 seems not to interact with the promoter of *OsNAP* or *OsNAC2* (Figure 6f).

Previous studies have reported that the microRNA osa-miR164b is closely associated with the post-transcriptional regulation of *OsNAC2*, resulting in reduction of *OsNAC2* mRNA levels [42,43]. To examine whether OsWRKY5 regulates endogenous levels of osa-miR164b during DIS, we determined the expression of osa-miR164b in detached leaves of three-week-old WT, *oswrky5-D*, and *oswrky5* plants using stem-loop RT-PCR analysis. It revealed no difference in the levels of osa-miR164b among genotypes (Appendix A), indicating that *OsWRKY5* seems not to participate in osa-miR164b-mediated senescence pathway.

### 2.5. OsWRKY5 Is Involved in ABA Biosynthesis Pathway

Among phytohormones affecting the onset and progression of leaf senescence [2], ABA activates senescence-associated regulatory pathways, leading to acceleration of leaf senescence [44]. Genetic studies have revealed that the endogenous ABA concentration is delicately controlled by senNACs such as OsNAP and OsNAC2 [25,40]. Considering that OsWRKY5 upregulated the expression of *OsNAP* and *OsNAC2*, we speculated that OsWRKY5 is mainly involved in regulating ABA biosynthesis. Indeed, the endogenous ABA concentration was significantly higher in leaves of three-week-old *oswrky5-D* plants than in those of the WT (Figure 7a). RT-qPCR analysis showed that ABA biosynthesis genes including *OsNCED3*, *OsNCED4*, and *OsNCED5* were significantly upregulated in *oswrky5-D* leaves compared with the WT (Figure 7b). This strongly suggests that the early leaf senescence of *oswrky5-D* is mainly due to an increase in ABA biosynthesis after heading.

To investigate whether phytohormones affect the expression of *OsWRKY5*, we next measured the expression of *OsWRKY5* in ten-day-old WT seedlings exogenously treated with epibrassinolide (BR), gibberellic acid (GA), indole-3-acetic acid (IAA), 6-benzylaminopurine (6-BA), salicylic acid (SA), methyl jasmonic acid (MeJA), ABA, or 1-aminocyclopropane-1-carboxylic acid (ACC). RT-qPCR showed that *OsWRKY5* expression was significantly reduced by MeJA and ABA treatments (Figure 7c), indicating that excessive levels of ABA decrease the expression of *OsWRKY5*.

## 3. Discussion

### 3.1. OsWRKY5 Promotes Leaf Yellowing during NS and DIS

We found that *OsWRKY5* participates in the ABA-mediated regulatory pathways of leaf senescence. *OsWRKY5* was expressed in leaves and its transcription was activated by aging and dark treatment (Figure 1b,c). In addition, we found four W-boxes and a single G-box in the 1500-bp upstream of transcription initiation site of *OsWRKY5* (Appendix A). These *cis*-elements are recognized by WRKY, bZIP, bHLH, and NAC transcription factors which are involved in the regulation of leaf senescence [28], implying that expression of *OsWRKY5* could be regulated by senescence-induced transcription factors. The WRKY domain of OsWRKY5 has the highest amino acid similarity to that of AtWRKY6 (Appendix A). Similar to the early leaf senescence phenotype of *AtWRKY6*-OX in Arabidopsis [29], the progression of leaf senescence was much faster in the *oswrky5-D* mutant than in WT plants under NS and DIS conditions. (Figure 3 and Figure 5), and the *oswrky5* knockdown mutant showed markedly delayed leaf senescence (Figure 3).

Many senescence-induced TFs directly or indirectly regulate expression of their target genes, including CDGs, SAGs, and other senescence-associated TFs. OsNAP directly binds to the promoters of *SGR*, *NYC1*, *NYC3*, *RCCR1*, and *Osl57* (encoding a putative 3-ketoacyl-CoA thiolase). OsNAP also indirectly regulates the expression of *Osh36* and *Osh69*, whose amino acid sequences are quite similar to those of *Arabidopsis thaliana* aminotransferase and *Brassica oleracea* seed imbibition protein, respectively [25]. OsNAC2 enhances chlorophyll degradation by directly interacting with the promoters of *SGR* and *NYC3* [40]. We therefore speculate that *OsWRKY5* upregulates the expression of CDGs and SAGs by regulating senNACs. Upregulation of *OsNAP* and *OsNAC2* was observed in *oswrky5-D*, resulting in early leaf yellowing (Figure 4 and Figure 5). However, OsWRKY5 does not bind to the promoter regions of *OsNAP* and *OsNAC2* despite the presence of repetitive TGAC core sequences (Figure 6e,f), suggesting that it indirectly regulates expression of these genes.

WRKY TFs can physically interact with other TFs involved in leaf senescence. For example, Besseau et al. (2012) showed that expression of Arabidopsis *WRKY30*, *WRKY53*, *WRKY54*, and *WRKY70* is induced during leaf senescence and WRKY53, WRKY54, and WRKY70 interact independently with WRKY30 in yeast two-hybrid assays [32]. In Arabidopsis, WRKY45 functions in GA-mediated leaf senescence by interacting with the DELLA protein RGA-LIKE1 (RGL1) characterized as a repressor of GA signaling [33]. Recently, TT2, a MYB family member, was identified as an interacting partner of WRKY27 in upland cotton (*Gossypium hirsutum* L.) [45]. Therefore, exploring the possible interaction networks of the OsWRKY5 TF in the regulation of senNACs should provide more insight into the mechanism of leaf senescence.

Interestingly, despite of constitutive expression of *OsWRKY5* in *oswrky5-D*, there were no significant difference in senescence phenotype and expression of CDGs and SAGs between WT and *oswrky5-D* before leaf senescence (Figure 3a and Figure 5a). Leaf senescence is a complex process involving numerous regulators [27,46]. These regulators are mostly induced by the onset of senescence. OsWRKY5 may require other cofactors to upregulate the expression levels of CDGs and SAGs during leaf senescence. Thus, even though *OsWRKY5* transcripts are highly accumulated in *oswrky5-D* during vegetative growth, it seems that *OsWRKY5* overexpression is not much as effective as a senescence promoter, possibly due to a lack of senescence-induced cofactors.

### 3.2. OsWRKY5 Mediates ABA-Induced Leaf Senescence

ABA promotes the onset and progression of leaf senescence [47]. Thus, endogenous ABA levels are elevated by upregulation of ABA biosynthesis genes during leaf senescence, promoting further ABA-induced leaf senescence [44,48]. Arabidopsis *9-CIS-EPOXYCAROTENOID DIOXYGENASE* (*NCED*) genes, encoding a rate-limiting enzyme in ABA biosynthesis, are upregulated during NS [19,49]. Dark incubation induces expression of *OsNCED3* in rice leaves [50], and overexpression of *OsNCED3* accelerates leaf yellowing in rice during dark incubation. *NAP* increases ABA biosynthesis by directly upregulating transcription of *ABSCISIC ALDEHYDE OXIDASE3* (*AAO3*), leading to chlorophyll degradation during dark incubation in Arabidopsis [20]. In rice, although transcription of ABA biosynthesis genes, such as *OsNCED1*, *OsNCED3*, *OsNCED4*, and *OsZEP*, is inhibited by OsNAP, the functional ortholog of Arabidopsis NAP, overexpression of *OsNAP* leads to precocious leaf senescence by directly regulating CDGs and SAGs [25]. *OsNAC2* elevates endogenous ABA content by directly binding to the promoters of *OsNCED3* and *OsZEP*, thereby promoting leaf senescence [40]. Transgenic Arabidopsis plants heterologously expressing foxtail millet (*Setaria italica*) *NAC1* (*SiNAC1*) show enhanced transcription of ABA biosynthesis genes, *NCED2* and *NCED3*, resulting in early leaf senescence [51]. Although ABA signaling pathways mediated by WRKY TFs are involved in multiple aspects of plant development including leaf senescence [52], molecular evidence for WRKY TF involvement in ABA biosynthesis is limited. In this study, we found that OsWRKY5 upregulates transcription of ABA biosynthesis genes, *OsNCED3*, *OsNCED4*, and *OsNCED5* (Figure 7b), suggesting that OsWRKY5 functions in the promotion of leaf senescence by increasing ABA biosynthesis (Figure 7a). Furthermore, based on the involvement of *OsWRKY5* in *OsNAC2* expression (Figure 6b,d), *OsWRKY5* probably activates an *OsNAC2*-mediated ABA biosynthetic pathway (Figure 8).

Plants have developed several regulatory mechanisms to restore ABA homeostasis during leaf senescence. For example, in tomato, *NAP2* directly regulates expression of genes regulating ABA biosynthesis (*NCED1*) and ABA degradation (*CYP707A2*) to establish ABA homeostasis during leaf senescence [53]. ABA-induced *OsNAP* represses the accumulation of endogenous ABA in rice, indicating that *OsNAP* participates in a negative feedback mechanism on ABA biosynthesis [25]. Expression of *OsNAC2* is differentially regulated by ABA concentration; *OsNAC2* expression is upregulated by 20 μM ABA, but inhibited by ABA concentrations over 40 μM [40]. Because the expression of *OsWRKY5* is reduced by excessive ABA treatment (Figure 7c), it is highly possible that *OsWRKY5* transcription is repressed by excessive concentrations of ABA via a negative feedback regulatory mechanism.

## 4. Materials and Methods

### 4.1. Plant Materials, Growth Conditions, and Experimental Treatments

The *Oryza sativa japonica* cultivar ‘Dongjin’ (parental line), and the *oswrky5-D* and *oswrky5* mutants were grown in a growth chamber under LD conditions (14 h light at 28 °C/10 h dark at 25 °C) and in a rice paddy field under NLD conditions (≥14 h sunlight/day, 37° N latitude) in Seoul, Republic of Korea. The T-DNA insertion activation-tagged *oswrky5-D* and knockdown *oswrky5* mutants were obtained from the Crop Biotech Institute at Kyung Hee University, Republic of Korea [54,55].

For dark treatment, detached leaves of rice plants grown in the growth chamber for 3 weeks were incubated in 3 mM 2-(N-morpholino)ethanesulfonic (MES) buffer (pH 5.8) with the abaxial side up at 28 °C in complete darkness. To detect *OsWRKY5* transcript levels under various hormone treatments, WT seeds were sterilized with 70% ethanol and 2% NaClO, and then germinated and grown on half-strength Murashige and Skoog (0.5X MS, Duchefa, The Netherlands) solid medium under LD conditions for 10 days in a growth chamber. Ten-day-old plants were transferred to 0.5X MS liquid medium containing 50 μM epibrassinolide (BR), 50 μM GA, 50 μM IAA, 50 μM 6-BA, 100 μM SA, 50 μM MeJA, 50 μM ABA, or 50 μM ACC. Total RNA was extracted from leaves harvested at 0 and 6 h of treatment.

### 4.2. Subcellular Localization

Full-length cDNA of *OsWRKY5* was amplified using gene-specific primers (Appendix A), cloned into pCR8/GW/TOPO vector (Invitrogen, Carlsbad, CA, USA), and then transferred into pEarleyGate104 (pEG104) gateway binary vector using Gateway LR clonase II enzyme mix (Invitrogen), resulting in a *35S::YFP-OsWRKY5* construct. The pEG104 vector and recombinant constructs were introduced into onion (*Allium cepa*) epidermal cells using a DNA particle delivery system (Biolistic PDS-1000/He, Bio-Rad, Hercules, CA, USA). After incubation at 25 °C for 16 h, green fluorescence was detected using a confocal laser scanning microscope (SP8X, Leica, Wetzlar, Germany). To visualize nuclei, samples were stained with 10 mL of 1 μg·mL^−1^ 4′,6-diamidino-2-phenylindole dihydrochloride (DAPI) dissolved in water for 10 min then viewed using a fluorescence microscope under ultraviolet light irradiation with appropriate filters.

### 4.3. Determination of Photosynthetic Activity, Total Chlorophyll, and SPAD Value

To evaluate photosynthetic activity, the middle section of the flag leaf of plants grown in a paddy field under NLD conditions was adapted in the dark for 10 min. The *Fv/Fm* ratio was then measured using an OS-30p+ instrument (Opti-Sciences, Hudson, NH, USA). Total chlorophyll content was measured in rice leaves grown in the growth chamber for 4 weeks. Pigment was extracted from detached leaves incubated in complete darkness using 80% ice-cold acetone. After centrifugation at 10,000× *g* for 15 min at 10 °C, the absorbance of supernatants was measured at 647 and 663 nm using a UV/VIS spectrophotometer (BioTek Instruments, Winooski, VT, USA). The concentration of chlorophyll was calculated as previously described [56]. The change of SPAD value was determined in the flag leaf of plants grown in a paddy field under NLD conditions using a SPAD-502 instrument (KONICA MINOLTA, Tokyo, Japan).

### 4.4. RT-qPCR and Stem-Loop RT-qPCR Analysis

Total RNA was extracted from rice tissues using an RNA Extraction kit (MG Med, Seoul, Korea), according to the manufacturer’s instructions. For synthesis of first-strand cDNA, 2 μg of total RNA was used for reverse transcription (RT) in 20 μL volume with oligo(dT)_15_ primer and M-MLV reverse transcriptase (Promega, Madison, WI, USA). For quantification of miR164b, stem-loop pulsed RT was conducted from 2 μg of total RNA in 20 μL volume using a miR164b-specific stem-loop primer and M-MLV reverse transcriptase (Promega) with the following conditions: 16 °C for 30 min followed by pulsed RT of 40 cycles at 16 °C for 2 min, 42 °C for 1 min, and 50 °C for 1 s, and then inactivation of reverse transcription at 70 °C for 5 min [57]. All RT products were diluted with 80 µL distilled water.

qPCR was performed with gene-specific primers and normalized to *UBIQUITIN5* (*UBQ5*) (Os01g22490) or rice U6 snRNA (Appendix A) according to the 2^−ΔΔ*C*t^ method [58]. The 20 µL reaction mixture included 2 µL cDNA from RT or stem-loop pulsed RT, 1 µL 0.5 µM primer, and 10 μL 2X GoTaq master mix (Promega). qPCR amplifications were conducted with a LightCycler 480 (Roche, Basel, Switzerland) using the following program: 94 °C for 2 min followed by 40 cycles of 94 °C for 15 s and 60 °C for 1 min.

### 4.5. Yeast One-Hybrid Assays

The coding sequence of *OsWRKY5* was amplified by PCR. The PCR product was subcloned into the *Eco*RI and *Pst*I sites of pGAD424 vector (Clontech, Kusatsu, Japan). Fragments of *OsNAP* and *OsNAC2* promoters containing the repetitive W-box core sequence (TGAC) were amplified by PCR and then cloned into pLacZi vector using *Eco*RI*-Xba*I, *Eco*RI*-Xba*I, *Sal*I*-Xho*I, *Sal*I*-Xho*I, *Sal*I*-Xho*I, *Sal*I*-Xho*I, and *Sal*I*-Xho*I sites, generating *OsNAP-1::LacZi*, *OsNAP-2::LacZi*, *OsNAC2-1::LacZi*, *OsNAC2-2::LacZi*, *OsNAC2-3::LacZi*, *OsNAC2-4::LacZi*, and *OsNAC2-5::LacZi* reporter constructs, respectively (Clontech, USA). These vectors and empty vector were transformed into yeast strain YM4271 using the PEG/LiAc method, and yeast cells were incubated in SD/-His/-Leu liquid medium. β-Galactosidase activity was determined by absorbance of chloramphenicol red, a hydrolysis product of chlorophenol red-β-D-galactopyranoside (CPRG), at 595 nm using a UV/VIS spectrophotometer (BioTek Instruments, Winooski, VT, USA) according to the Yeast Protocol Handbook (Clontech).

### 4.6. Determination of ABA Content

Four-week-old leaves of WT and *oswrky5-D* were pulverized in liquid nitrogen and then homogenized in 80% methanol containing 1 mM butylated hydroxytoluene as an antioxidant. Extracts incubated for 12 h at 4 °C were centrifuged at 4000× *g* for 20 min. The supernatant was passed through a Sep-Pak C18 cartridge (Waters, Milford, MA, USA) as described previously [59], and the eluate was subjected to an enzyme-linked immunosorbent assay (ELISA) using an ABA ELISA kit (MyBioSource, San Diego, CA, USA) according to the manufacturer’s instructions.

## 5. Conclusions

We investigated a WRKY TF family member in rice, OsWRKY5, which acts as a positive regulator of leaf senescence. OsWRKY5 upregulates both ABA biosynthesis and transcription of CDGs and SAGs during leaf senescence, thereby promoting leaf yellowing. *OsWRKY5* transcription contributes to regulating leaf senescence in rice.

## Figures and Tables

**Figure 1 ijms-20-04437-f001:**
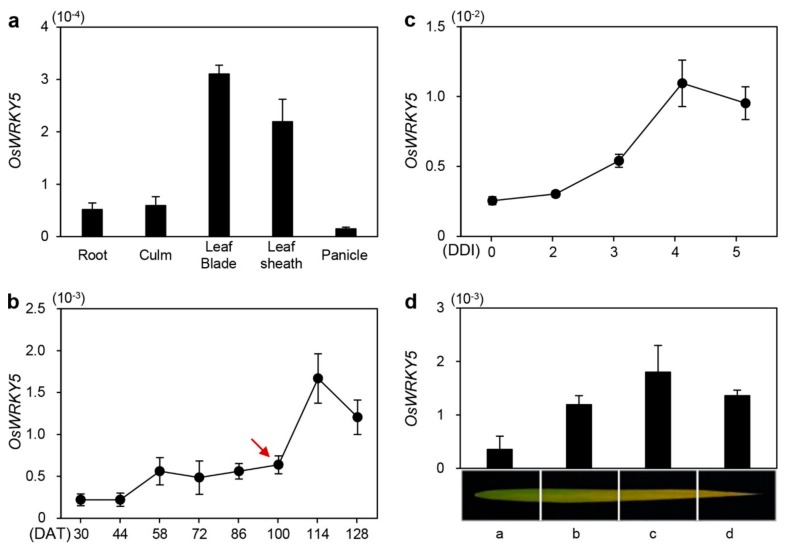
Expression profiles of *OsWRKY5* in rice. (**a**) *OsWRKY5* mRNA levels in detached organs from the *japonica* cultivar ‘Dongjin’ (hereafter wild type; WT) at the heading stage. *OsWRKY5* was mainly expressed in leaf blade and leaf sheath. (**b**,**c**) Changes in *OsWRKY5* expression level in leaf blades of WT rice grown in a paddy field (**b**) or in the greenhouse (**c**) under natural long day conditions (≥14 h light/day). (**c**) Detached leaves of three-week-old WT were incubated in 3 Mm 2-(N-morpholino)ethanesulfonic (MES) buffer (pH 5.8) at 28 °C in complete darkness. Red arrow indicates heading date. (**d**) Expression of *OsWRKY5* measured in flag leaves divided into four regions from the green sector (a) to the yellow sector (d) at 128 days after transplanting (DAT). *OsWRKY5* mRNA levels were determined by RT-qPCR analysis and normalized to that of *OsUBQ5* (Os01g22490). Mean and SD values were obtained from at least three biological samples. Experiments were repeated twice with similar results. DDI, day(s) of dark incubation.

**Figure 2 ijms-20-04437-f002:**
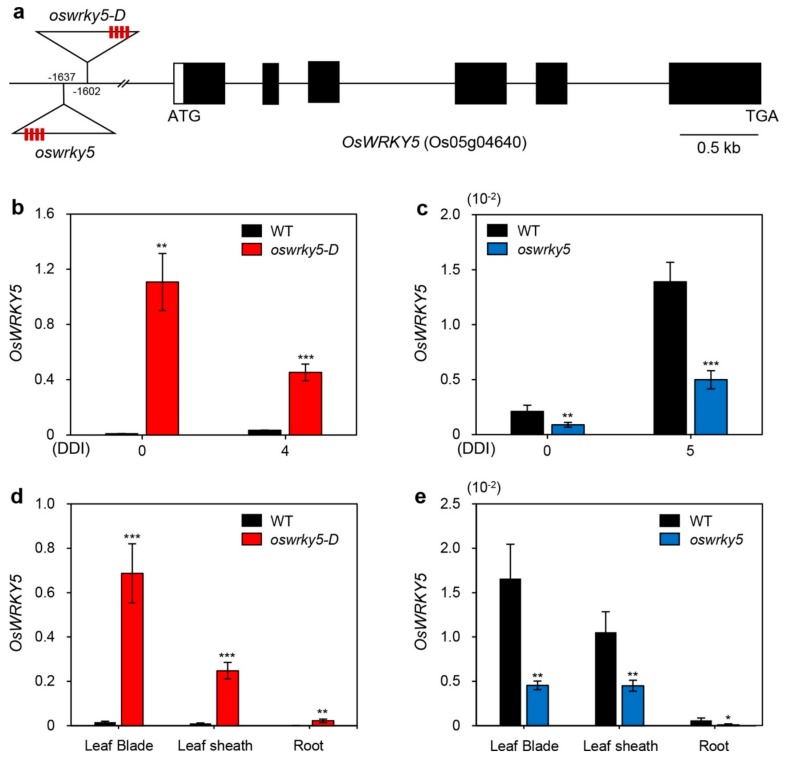
Mutation of *OsWRKY5* by T-DNA insertion. (**a**) Schematic diagram depicting the positions of T-DNA insertions in the promoter region of *OsWRKY5* (LOC_Os05g04640). Black and white bars represent exons and 5′-untranslated region, respectively. Open triangles indicate the location of the *OsWRKY5* T-DNA insertions (*oswrky5-D*, PFG_3A-15928; *oswrky5*, PFG_3A-06060). Red boxes on triangles represent tetramerized 35S enhancers (4 × 35S). (**b**,**c**) Total RNA was isolated from detached leaves of WT and mutant lines (*oswrky5-D* and *oswrky5*) under DIS as shown in Figure 3a,b. (**d**,**e**) *OsWRKY5* mRNA levels were measured in rice tissues separated from three-week-old WT and mutant lines. Transcript levels of *OsWRKY5* in *oswrky5-D* (**b**,**d**) and *oswrky5* (**c**,**e**) were determined by RT-qPCR and normalized to the transcript levels of *OsUBQ5*. Mean and SD values were obtained from more than three biological replicates. Asterisks indicate a statistically significant difference from WT, as determined by Student’s *t*-test (* *p* < 0.05, ** *p* < 0.01, *** *p* < 0.001). DDI, day(s) of dark incubation.

**Figure 3 ijms-20-04437-f003:**
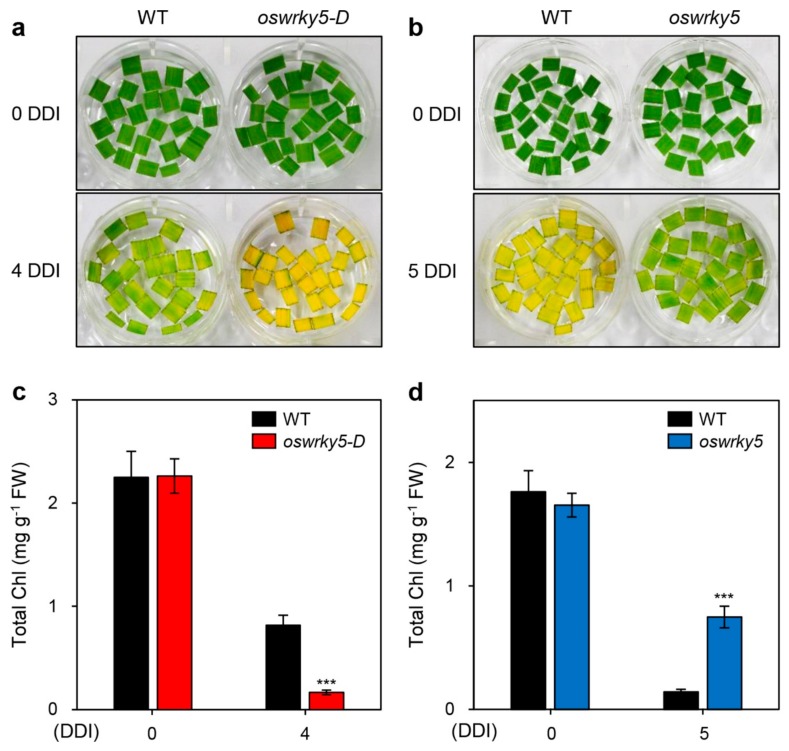
*OsWRKY5* promotes leaf yellowing under dark-induced senescence (DIS) conditions. WT and mutant lines (*oswrky5-D* and *oswrky5*) were grown in paddy soil for four weeks under natural long day conditions (≥14 h light/day). (**a**,**b**) Yellowing of detached leaves induced in 3 mM MES buffer (pH 5.8) at 28 °C under complete darkness. Changes in leaf color (**a**,**b**) and total chlorophyll (Chl) contents (**c**,**d**) of *oswrky5-D* or *oswrky5* mutants compared with the WT after 4 or 5 days of dark incubation (DDI), respectively. Mean and SD values were obtained from more than three biological replicates. Asterisks indicate a statistically significant difference from WT, as determined by Student’s *t*-test (*** *p* < 0.001). Experiments were repeated twice with similar results. FW, fresh weight.

**Figure 4 ijms-20-04437-f004:**
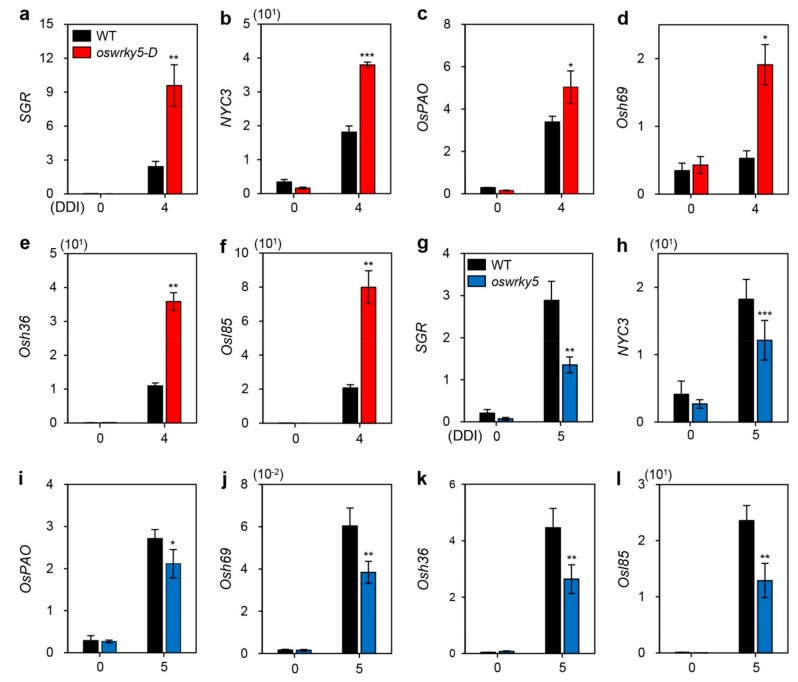
Altered expression of CDGs and SAGs in *oswrky5-D* and *oswrky5* during DIS. Total RNA was isolated from detached leaves of WT and mutant lines (*oswrky5-D* and *oswrky5*) under DIS as shown in Figure 3 (3a,b). Expression of CDGs and SAGs in *oswrky5-D* (**a**–**f**) or *oswrky5* (**g**–**l**) was compared with that in the WT after 4 or 5 DDI, respectively. Transcript levels of CDGs (**a**–**c** and **g**–**i**) and SAGs (**d**–**f** and **j**–**l**) were determined by RT-qPCR analysis and normalized to that of *OsUBQ5*. Mean and SD values were obtained from more than three biological replicates. Asterisks indicate a statistically significant difference from WT, as determined by Student’s *t*-test (* *p* < 0.05, ** *p* < 0.01, *** *p* < 0.001). Experiments were repeated twice with similar results. CDGs, Chl degradation genes; DDI, day(s) of dark incubation; SAGs, senescence-associated genes.

**Figure 5 ijms-20-04437-f005:**
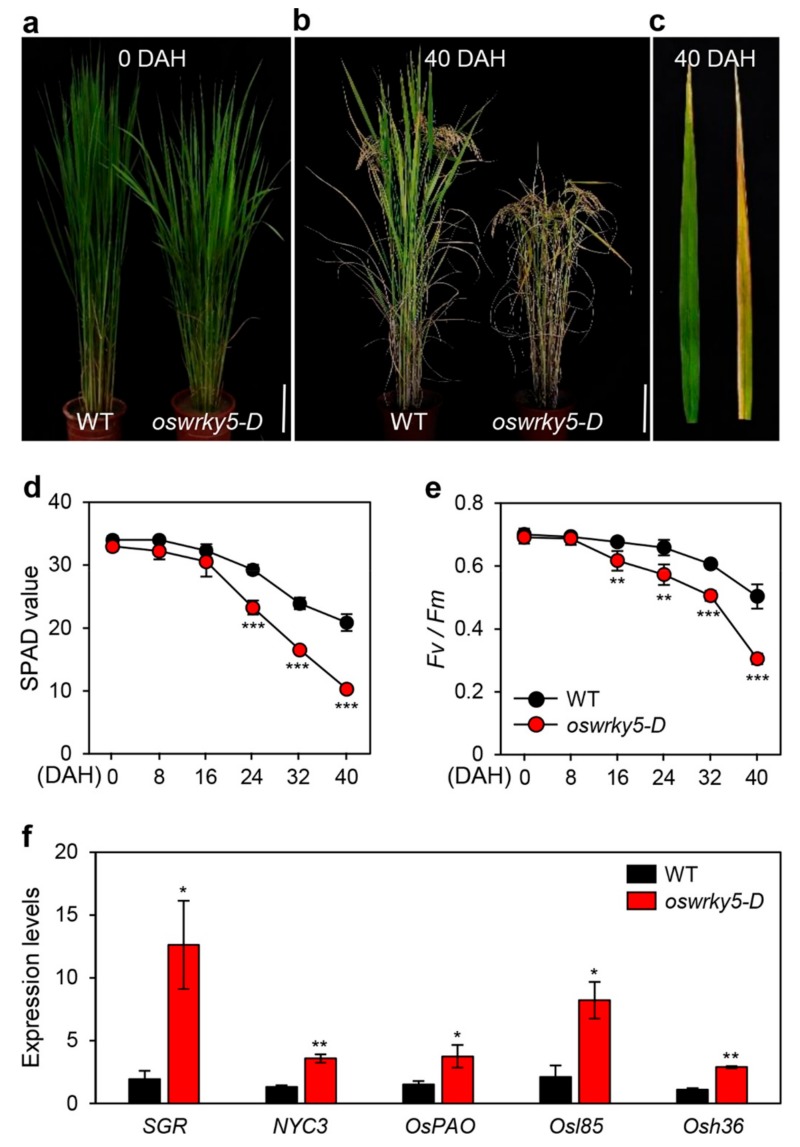
*oswrky5-D* promotes leaf senescence during NS. WT and *oswrky5-D* plants were grown in a paddy field under natural long-day conditions (≥14 h light/day). (**a,b**) Phenotypes of WT and *oswrky5-D* plants at heading (0 DAH) (**a**) and 40 days after heading (DAH) (**b**). White scale bars = 20 cm. (**c**) Senescing flag leaves of WT (*left*) and *oswrky5-D* (*right*) at 40 DAH. Photos shown are representative of five independent plants. (**d**–**e**) Changes in SPAD value (**d**) and photosystem II (PSII) activity (*Fv/Fm*) (**e**) in flag leaves at heading. (**f**) Expression of CDGs and SAGs measured in senescing flag leaves (**c**). Transcript levels were determined by RT-qPCR analysis and normalized to that of *OsUBQ5*. Mean and SD values were obtained from more than three biological replicates. Asterisks indicate a statistically significant difference from WT, as determined by Student’s *t*-test (* *p* < 0.05, ** *p* < 0.01, *** *p* < 0.001).

**Figure 6 ijms-20-04437-f006:**
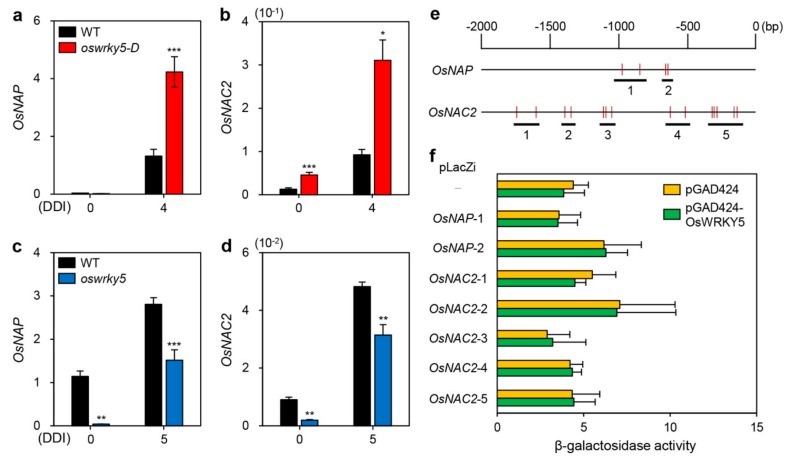
OsWRKY5 indirectly regulates expression of senescence-induced NAC TFs. (**a**–**d**) Total RNA was isolated from detached leaves of WT and mutant lines (*oswrky5-D* and *oswrky5*), as shown in Figure 3 (a,b). Transcript levels of *OsNAP* (**a**,**c**) and *OsNAC2* (**b**,**d**) were determined by RT-qPCR analysis and normalized to the transcript levels of *OsUBQ5*. Mean and SD values were obtained from more than three biological replicates. Asterisks indicate a statistically significant difference from WT, as determined by Student’s *t*-test (* *p* < 0.05, ** *p* < 0.01, *** *p* < 0.001). (**e**,**f**) Interaction of OsWRKY5 with the promoters of *OsNAP* and *OsNAC2* by yeast one-hybrid assays. (**e**) Numbers represent upstream base pairs from the transcriptional initiation sites of *OsNAP* and *OsNAC2*. Vertical red lines represent the W-box core sequence (TGAC). Horizontal black bars represent regions containing repetitive TGAC sequences. (**f**) β-Galactosidase activity of bait plasmids (pGAD424 and pGAD424-OsWRKY5) evaluated by the absorbance of chloramphenicol red, a hydrolysis product of chlorophenol red-β-D-galactopyranoside (CPRG). Empty bait (pGAD424) and prey plasmids (-) were used for negative controls. Experiments were repeated twice with similar results. DDI, day(s) of dark incubation.

**Figure 7 ijms-20-04437-f007:**
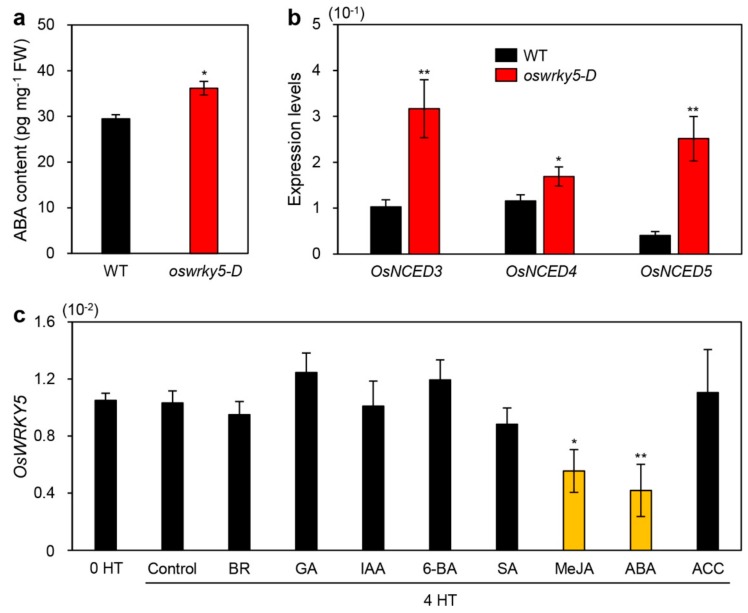
*OsWRKY5* participates in ABA-mediated senescence pathways. (**a**) Endogenous ABA contents measured in leaves of WT and *oswrky5-D* plants grown in paddy soil for 3 weeks under LD conditions. FW, fresh weight. (**b**) Total RNA was extracted from leaves of the same WT and *oswrky5-D* plants used for the analysis shown in Figure 6A. Transcript levels of ABA biosynthetic genes including *OsNCED3*, *OsNCED4*, and *OsNCED5* were determined by RT-qPCR analysis and normalized to transcript levels of *OsUBQ5*. Mean and SD values were obtained from more than three biological replicates. Asterisks indicate a statistically significant difference from WT, as determined by Student’s *t*-test (* *p* < 0.05, ** *p* < 0.01). (**c**) Ten-day-old WT seedlings grown on 0.5X MS phytoagar medium at 28 °C under continuous light conditions were transferred to 0.5X MS liquid medium only (control) or 0.5X MS liquid medium containing 50 μM epibrassinolide (BR), 50 μM gibberellic acid (GA), 50 μM 3-indoleacetic acid (IAA), 50 μM 6-benzylaminopurine (6-BA), 100 μM salicylic acid (SA), 50 μM methyl jasmonic acid (MeJA), 50 μM abscisic acid (ABA), or 50 μM 1-aminocyclopropane-1-carboxylic acid (ACC). Total RNA was isolated from leaves after 4 h of treatment. *OsWRKY5* mRNA levels were determined by RT-qPCR analysis and normalized to transcript levels of *OsUBQ5*. Mean and SD values were obtained from more than three biological replicates. Asterisks on orange bars indicate a statistically significant difference from the control, as determined by Student’s *t*-test (**p* < 0.05, ** *p* < 0.01). Experiments were repeated twice with similar results.

**Figure 8 ijms-20-04437-f008:**
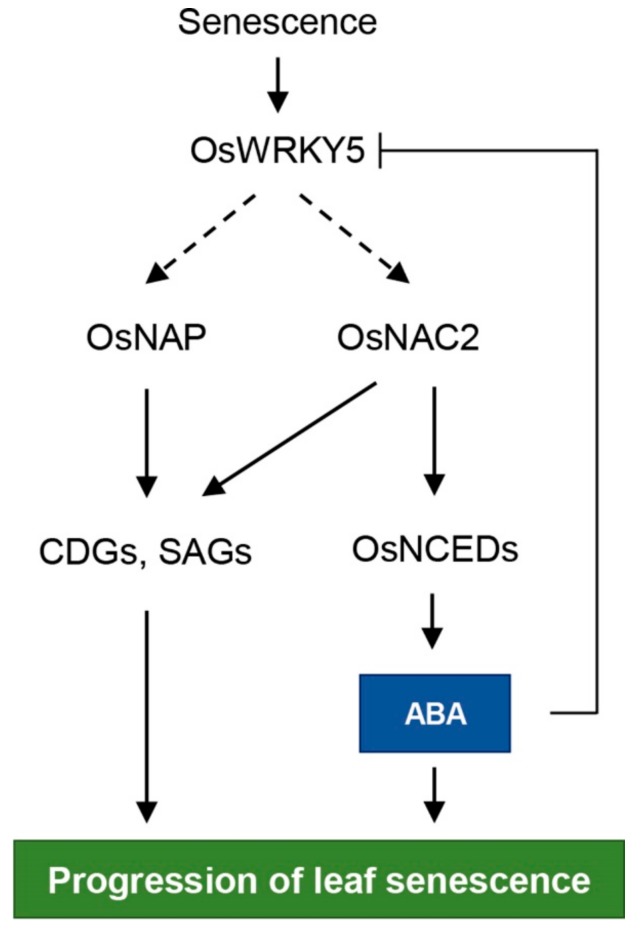
Proposed model for the role of OsWRKY5 in leaf senescence. Arrows indicate activation and bar-ended line represents inhibition. Solid and dashed arrows represent direct and indirect regulation, respectively.

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
