# Peer review of "OsWRKY5 Promotes Rice Leaf Senescence via Senescence-Associated NAC and Abscisic Acid Biosynthesis Pathway"

_ijms, 2019, doi:10.3390/ijms20184437_

Round 1

Reviewer 1 Report

This manuscript revealed that the OsWRKY5 promoted the leaf senescence, contributing to elucidate the molecular mechanism of rice leaf senescence. However, there are several minor comments to improve the manuscript.

(1)Figure 1 legend

(b, c) Changes in OsWRKY5 expression level in leaf blades of WT rice grown in a paddy field (b) or in the greenhouse (c) under natural long day conditions (≥14 h light/day).

Figure 1c represents the OsWRKY5 expression levels in the detached leaves during DIS (Page 3; line 112-114). However, the legend of Figure 1c does not have any information about that the leaves are detached and incubated in dark conditions. Additional description should be mentioned in Figure1c legend.

(2) (Page 9; line 221-224)RT-qPCR showed that compared with the WT, in oswrky5 the expression levels of OsNAP and OsNAC2 were higher during dark incubation (Figure 6a,b), while they were downregulated at 0 and 4 DDI compared with the WT (Figure 6c,d). 

…should be

RT-qPCR showed that compared with the WT, in “oswrky5-D” the expression levels of OsNAP and OsNAC2 were higher during dark incubation (Figure 6a,b), while “in oswrky5” they were downregulated at 0 and "45” DDI compared with the WT (Figure 6c,d) ?  

(3) Until heading, WT and oswrky5-D had no significant differences in leaf color (Figure 5a). On the other hand, the leaves from young plant (three-week-old) in oswrky5-D showed higher expression levels of OsWRKY5 (Figure2d). Does rice plant have a mechanism that prevent the promotion of leaf senescence by OsWRKY5 before the heading stage?

Additionally, the expression levels of CDGs and SAGs did not have differences before the dark incubation (0 DDI) between WT and oswrky5-D (Figure 4).

It would be better to discuss why the leaf senescence was not promoted by OsWRKY5 before the heading stage or dark incubation.

Author Response

This manuscript revealed that the OsWRKY5 promoted the leaf senescence, contributing to elucidate the molecular mechanism of rice leaf senescence. However, there are several minor comments to improve the manuscript.

Q1, Figure 1 legend

(b, c) Changes in OsWRKY5 expression level in leaf blades of WT rice grown in a paddy field (b) or in the greenhouse (c) under natural long day conditions (≥14 h light/day).

Figure 1c represents the OsWRKY5 expression levels in the detached leaves during DIS (Page 3; line 112-114). However, the legend of Figure 1c does not have any information about that the leaves are detached and incubated in dark conditions. Additional description should be mentioned in Figure 1c legend.

Answer 1:

Thank you for pointing out our mistake. We added the description in the legend of Figure 1c: “(c) Detached leaves of three-week-old WT were incubated in 3 mM MES buffer (pH 5.8) at 28°C in complete darkness”. (P3, L119-120)

Q2. (Page 9; line 221-224) RT-qPCR showed that compared with the WT, in oswrky5 the expression levels of OsNAP and OsNAC2 were higher during dark incubation (Figure 6a,b), while they were downregulated at 0 and 4 DDI compared with the WT (Figure 6c,d).

…should be

RT-qPCR showed that compared with the WT, in “oswrky5-D” the expression levels of OsNAP and OsNAC2 were higher during dark incubation (Figure 6a,b), while “in oswrky5” they were downregulated at 0 and "45” DDI compared with the WT (Figure 6c,d) ?

Answer 2:

Thank you for correcting our mistake. As the reviewer’s suggestion, we revised this sentence: “RT-qPCR showed that compared with WT, the expression levels of OsNAP and OsNAC2 were higher in oswrky5-D during dark incubation (Figure 6a,b), while they were downregulated at 0 and 5 DDI in oswrky5 compared with the WT (Figure 6c,d). These results suggest that OsWRKY5 promote leaf senescence by upregulating the expression of OsNAP and OsNAC2.” (P9, L218-222)

Q3. Until heading, WT and oswrky5-D had no significant differences in leaf color (Figure 5a). On the other hand, the leaves from young plant (three-week-old) in oswrky5-D showed higher expression levels of OsWRKY5 (Figure2d). Does rice plant have a mechanism that prevent the promotion of leaf senescence by OsWRKY5 before the heading stage?

Additionally, the expression levels of CDGs and SAGs did not have differences before the dark incubation (0 DDI) between WT and oswrky5-D (Figure 4).

It would be better to discuss why the leaf senescence was not promoted by OsWRKY5 before the heading stage or dark incubation.

Answer 3:

Thank you for your valuable comment. We added the information in the Discussion section (P11, L326-331). - “Leaf senescence is a complex process involving numerous regulators [47, 48]. These regulators are mostly induced by the onset of senescence. OsWRKY5 may require other cofactors to upregulate the expression of CDGs and SAGs during leaf senescence. Thus, even though OsWRKY5 transcripts are highly accumulated in oswrky5-D during vegetative growth, it seems that OsWRKY5 overexpression is not much as effective as a senescence promoter, possibly due to a lack of senescence-induced cofactors”.

Reviewer 2 Report

The manuscript characterized transcription factor OsWRKY5 and its function in leaf senescence through analysis on two mutants (overexpression and knockdown). It concluded that “OsWRKY5 is a positive regulator of leaf senescence that upregulates senescence-induced NAC genes leading to expression of ABA biosynthesis and chlorophyll degradation genes”. The manuscript was well written and the data was well presented, but some conclusions were not appropriate. Here are some concerns:

Introduction: Some description in paragraph 1 and 3 is kind of repetitive. Line 224-245 “These results suggest that “OsWRKY5 acts upstream of the OsNAP and OsNAC2 regulatory pathways to promote leaf senescence.” In my opinion, the conclusion is not convincible based on (only) RT-PCR data. More explanation may be required. Similarly, Line 242-243 “OsWRKY5 indirectly regulates OsNAC2 independent of osa-miR164b.” Whether the conclusion is appropriate is questioned. The big concern from my side is the data shown in Fig. 2 and Fig.4. Fig2(b,c) showed the expression of OsWRKY5 in the two mutants during DIS. Obviously, the expression of OsWRKY5 in overexpression line was much higher than WT. However, its expression largely decreased in oswrky5-D and increased in oswrky5 during DIS (from 0-4 DDI), while its expression showed increase for WT. Fig.4 showed that several genes was upregulated in oswrky5-D but downregulated in oswrky5 during DIS (0-4 DDI), and the conclusion was given that “OsWRKY5 promotes the onset and progression of leaf senescence by upregulating expression of CDGs and SAGs.” In my opinion, since the expression of OsWRKY5 decreased in oswrky5-D during DIS, the expression of genes regulated by OsWRKY5 might also be decreased but not largely increased, even though their expression was still higher than that in WT. Similar situation applied to the knockdown line. Therefore, whether those genes are indeed regulated by OsWRKY5 is confusing. At least, clear explanation is required.

Author Response

The manuscript characterized transcription factor OsWRKY5 and its function in leaf senescence through analysis on two mutants (overexpression and knockdown). It concluded that “OsWRKY5 is a positive regulator of leaf senescence that upregulates senescence-induced NAC genes leading to expression of ABA biosynthesis and chlorophyll degradation genes”. The manuscript was well written and the data was well presented, but some conclusions were not appropriate. Here are some concerns:

Q1. Introduction: Some description in paragraph 1 and 3 is kind of repetitive.

Answer 1:

We deleted the first sentence in paragraph 3 that seems to be repetitive. (P2, L51)

Q2. Line 224-245 “These results suggest that “OsWRKY5 acts upstream of the OsNAP and OsNAC2 regulatory pathways to promote leaf senescence.” In my opinion, the conclusion is not convincible based on (only) RT-PCR data. More explanation may be required. Similarly, Line 242-243 “OsWRKY5 indirectly regulates OsNAC2 independent of osa-miR164b.” Whether the conclusion is appropriate is questioned.

Answer 2:

We agree your opinions. We revised the sentences to clarify the conclusions:

“These results suggest that OsWRKY5 promote leaf senescence by upregulating the expression of OsNAP and OsNAC2” (P9, L221-222)

“, indicating that OsWRKY5 seems not to participate in osa-miR164b-mediated senescence pathway” (P9, L239-240)

Q3. The big concern from my side is the data shown in Fig. 2 and Fig.4. Figure 2 (b,c) showed the expression of OsWRKY5 in the two mutants during DIS. Obviously, the expression of OsWRKY5 in overexpression line was much higher than WT. However, its expression largely decreased in oswrky5-D and increased in oswrky5 during DIS (from 0-4 DDI), while its expression showed increase for WT. Fig.4 showed that several genes was upregulated in oswrky5-D but downregulated in oswrky5 during DIS (0-4 DDI), and the conclusion was given that “OsWRKY5 promotes the onset and progression of leaf senescence by upregulating expression of CDGs and SAGs.” In my opinion, since the expression of OsWRKY5 decreased in oswrky5-D during DIS, the expression of genes regulated by OsWRKY5 might also be decreased but not largely increased, even though their expression was still higher than that in WT. Similar situation applied to the knockdown line. Therefore, whether those genes are indeed regulated by OsWRKY5 is confusing. At least, clear explanation is required.

Answer 3:

Thank you for your comments. We realized that the Figure 2b could make the readers confused. To be honest, we have no answer why the transcriptional activity of 4X 35S enhancers (activation tagging) in the promoter of OsWRKY5 was reduced at 4 DDI (senescing yellow leaves) compared with at 0 DDI (non-senescing green leaves), while the expression of OsWRKY5 was increased by the native promoter of OsWRKY5 in WT during DIS. However, it is obvious that the expression levels of OsWRKY5 in oswrky5-D were much higher than that in WT at both 0 DDI and 4 DDI, although it appears that the expression of OsWRKY5 increased in WT and decreased in oswrky5-D during DIS (Figure 2b). Please excuse our insufficient knowledge and explanation regarding the expression activity of 4X 35S enhancers (activation tagging) in senescing yellow leaf tissues compared to in non-senescent green leaf tissues of rice.

Reviewer 3 Report

Kim et al. report the involvement of OsWRKY5 in the onset of leaf senescence. Because OsWRKY5 was upregulated during aging, the authors presumed that OsWRKY5 could be a regulator of the onset of leaf senescence. Using oswrky5-d and oswarky5, up- and down-regulated mutants, respectively, the authors tested the possibility of roles of OsWRKY5 in leaf senescence.

Overall, the topic would be interesting for potential readers. The manuscript is well written and data are clearly presented with statistical tests. In contrast, the reviewer concerns that the description on oswrky5-d and oswarky5 is insufficient to specifically define the genetic characters of these mutants. It is of note that two T-DNA activation tag insertions in the 5’ region of OsWRKY5 could activate and repress expression of the same gene. Ideally, backcrossing these two mutant to WT would demonstrate effects of these two insertions on the phenotype. In addition, generation of transgenic OsWRKY5 overexpressor and genome-editing plants could strengthen the conclusion. The reviewer understands that time limitation might hamper the authors’ willing to conduct such genetic analyses, but suggests the importance to identify the position where the insertions are located. Have the authors checked how many base pairs from the initiation codon of OsWRKY5 to the T-DNAs? Have they found possible DNA-binding motifs to explain up or down regulation of the WRKY gene? Database may provide some information, but it is definitely important for the authors to experimentally determine the insertion sites by themselves in the scientific report.

Minor comments;

Tile; Because the manuscript presents no evidence on the direct regulation of WRKY, NAC and ABA, the word “by” is misled.

Line 28; a negative feedback is not supported by data.

Line 30; it would be better to carefully describe NAC, ABA, and Chl. No data presents one gene leads to others.

Line 83; dark-induced senescence

Line 92-94; It seems likely that the genetic structure is derived from a database or report. Reference is needed.

Line 126; Specify the biological replicate in each experiment (here and in other figures).

Line 160; associated with upregulation of CDGs

Line 199; Fig. 3 (a, b)

Line 236; OsWRKY seems not to bind

Line 242; indirectly and independent. Please use a plain logic and reword the sentence.

Line 247; Fig. 3 (a, b)

Line 259 and 275; a negative feedback is not based on the data

Line 434; WRKY5 in ABA-induced senescence were not investigated.

Line 436; no negative feedback is appeared here.

Line 437; ABA homeostasis is not presented.

Author Response

Kim et al. report the involvement of OsWRKY5 in the onset of leaf senescence. Because OsWRKY5 was upregulated during aging, the authors presumed that OsWRKY5 could be a regulator of the onset of leaf senescence. Using oswrky5-d and oswarky5, up- and down-regulated mutants, respectively, the authors tested the possibility of roles of OsWRKY5 in leaf senescence.

Overall, the topic would be interesting for potential readers. The manuscript is well written and data are clearly presented with statistical tests. In contrast, the reviewer concerns that the description on oswrky5-d and oswrky5 is insufficient to specifically define the genetic characters of these mutants. It is of note that two T-DNA activation tag insertions in the 5’ region of OsWRKY5 could activate and repress expression of the same gene. Ideally, backcrossing these two mutant to WT would demonstrate effects of these two insertions on the phenotype. In addition, generation of transgenic OsWRKY5 overexpressor and genome-editing plants could strengthen the conclusion. The reviewer understands that time limitation might hamper the authors’ willing to conduct such genetic analyses, but suggests the importance to identify the position where the insertions are located.

Q1. Have the authors checked how many base pairs from the initiation codon of OsWRKY5 to the T-DNAs?

Answer 1.

Nucleotide sequences used for determining the locations of T-DNA insertions have been posted on the RiceGE database (http://signal.salk.edu/cgi-bin/RiceGE). Based on comparison with OsWRKY5 promoter sequence, we predicted that T-DNA fragments of oswrky5-D and oswrky5 are inserted in the 1602-bp and 1637-bp upstream of the start codon of OsWRKY5 gene, respectively (Figure 2a) (P5). It was confirmed by our re-sequencing the regions of T-DNA insertion regions of two mutant lines.

Q2. Have they found possible DNA-binding motifs to explain up or down regulation of the WRKY gene? Database may provide some information, but it is definitely important for the authors to experimentally determine the insertion sites by themselves in the scientific report.

Answer 2.

We found four W-boxes and a single G-box in the 1500-bp upstream of the transcription initiation site of OsWRKY5. These cis-elements are recognized by WRKY, bZIP, bHLH, and NAC transcription factors which are involved in the regulation of leaf senescence [45]. We added this information in the Discussion section (P11, L295-299) and Figure S4.

Minor comments;

Title; Because the manuscript presents no evidence on the direct regulation of WRKY, NAC and ABA, the word “by” is misled. - The title is revised as your comments: OsWRKY5 Promotes Rice Leaf Senescence via Senescence-Associated NAC and Abscisic Acid Biosynthesis Pathways. (P1, L2)

Line 28; a negative feedback is not supported by data. - Thank you for your notice. It is removed. (P1, L28)

Line 30; it would be better to carefully describe NAC, ABA, and Chl. No data presents one gene leads to others. - Thanks you for your comments. It is revised. (P1, L28-30)

Line 83; dark-induced senescence - It is corrected to DIS. (P2, L81)

Line 92-94; It seems likely that the genetic structure is derived from a database or report. Reference is needed. - We added the website of Rice Genome Annotation Project. (P2, L91)

Line 126; Specify the biological replicate in each experiment (here and in other figures). - We already mentioned the biological repeats in all the Figure legends. Mean and SD values were obtained from more than three biological replicates.

Line 160; associated with upregulation of CDGs - It is added. (P4, L151)

Line 199; Fig. 3 (a, b) - It is added. (P7, L195)

Line 236; OsWRKY seems not to bind - It is revised. (P9, L233)

Line 242; indirectly and independent. Please use a plain logic and reword the sentence. - Thanks for your comment. It is revised. (P9, L238-240)

Line 247; Fig. 3 (a, b) - It is added. (P9, L244)

Line 259 and 275; a negative feedback is not based on the data - It is revised. (P10, L256 and 272)

Line 434; WRKY5 in ABA-induced senescence were not investigated. – The paragraph in the Conclusions is revised. (P15, L440-443)

Line 436; no negative feedback is appeared here. - The paragraph in the Conclusions is revised. (P15, L440-443)

Line 437; ABA homeostasis is not presented. - The paragraph in the Conclusions is revised. (P15, L440-443)

Round 2

Reviewer 2 Report

Authors have revised based on reviewers' suggestions, and the manuscript is indeed much better than the previous version. In my opinion, however, the manuscript can be further improved, especially data analysis and discussion.

Author Response

Authors have revised based on reviewers' suggestions, and the manuscript is indeed much better than the previous version. In my opinion, however, the manuscript can be further improved, especially data analysis and discussion.

Answer: We did our best to clarify the biological function of OsWRKY5 in the regulation of rice leaf senescence in both Discussion and Conclusions sections. Please excuse us not fully enough to satisfy your opinion.

Reviewer 3 Report

The manuscript is appropriately revised. To more clearly explain genetic traits of the rice plants (Fig 2), it could be better to present answer 1 (authors' response) in the main text.

Author Response

The manuscript is appropriately revised. To more clearly explain genetic traits of the rice plants (Fig 2), it could be better to present answer 1 (authors' response) in the main text.

Answer: As the reviewer’s suggestion, we added the information of T-DNA insertion mutants in the Results section. (P4, L131-135)